# Retroactivity induced operating regime transition in an enzymatic futile cycle

**Akshay Parundekar, Ganesh A. Viswanathan** *

Department of Chemical Engineering, Indian Institute of Technology Bombay, Powai, Mumbai, India

* ganeshav@iitb.ac.in

## Abstract

Activated phosphorylation-dephosphorylation biochemical reaction cycles are a class of enzymatic futile cycles. A futile cycle such as a single MAPK cascade governed by two underlying enzymatic reactions permits Hyperbolic (H), Signal transducing (ST), Threshold-hyperbolic (TH) and Ultrasensitive (U) operating regimes that characterize input-output behaviour. Retroactive signalling caused by load due to sequestration of phosphorylated or unphosphorylated form of the substrate in a single enzymatic cascade without explicit feedback can introduce two-way communication, a feature not possible otherwise. We systematically characterize the operating regimes of a futile cycle subject to retroactivity in either of the substrate forms. We demonstrate that increasing retroactivity strength, which quantifies the downstream load, can trigger five possible regime transitions. Retroactivity strength is a reflection of the fraction of the substrate sequestered by its downstream target. Remarkably, the minimum required retroactivity strength to evidence any sequestration triggered regime transition demands 23% of the substrate bound to its downstream target. This minimum retroactivity strength corresponds to the transition of the dose-response curve from ST to H regime. We show that modulation of the saturation and unsaturation levels of the enzymatic reactions by retroactivity is the fundamental mechanism governing operating regime transition.

## 1. Introduction

Enzymatic cascades or futile cycles consisting of phosphorylation-dephosphorylation biochemical reaction cycles, are crucial, ubiquitously conserved, building-blocks of cellular signalling networks [1, 2]. An enzymatic futile cycle employs phosphorylation and dephosphorylation reactions, respectively catalysed by kinase and phosphatase, to enable transition of a protein substrate between its two forms, namely inactive and active. Enzymatic cascades impart important properties like responsiveness, robustness, specificity onto a signalling response [3, 4], weak signal amplification [5], signal speed acceleration [6], filter out noise in signal [7–9]. One such well-known enzymatic cascade is the Raf/MEK/ERK MAPK cascade, a key signal amplifier and a modulator of pro-survival and pro-apoptotic signalling pathways [10–12]. Aberrant functioning of this cascade has been implicated in many diseases such as cancer [13, 14]. Detailed understanding of the sustained and transient activation patterns of

**Funding:** This study was supported by Science and Engineering Research Board, Department of Science and Technology, Government of India (MTR/2020/000589 and CRG/2020/002672) for funding this study. The funders had no role in study design, data collection and analysis, decision to publish, or preparation of the manuscript.

**Competing interests:** The authors have declared that no competing interests exist.

MAPK cascade can therefore offer useful insights in designing therapeutic strategies for combating certain diseases.

Activation behaviour of a futile cycle as a response to a stimulus of certain strength and their dynamic evolution have traditionally been characterized by systematically studying the dose-response curves permitted by the cascade [15–17]. Dose-response curve or the input-output characteristic of the cycle at steady-state is a map of the abundances of the input kinase and of the active protein (output) of the cascade [18]. Based on the qualitative nature of the dose-response curve, dictated by the saturated/unsaturated state of the two enzymatic reactions, the activation behaviour of futile cycle have been classified into four distinct operating regimes, *viz.*, Hyperbolic (H), Signal transducing (ST), Threshold-hyperbolic (TH), Ultrasensitive (U), each of which display different signal processing capabilities [18, 19]. Operating regimes of the MAPK cascades juxtaposed with patient-stratification data have recently been considered in disease prognostics [20]. Recently, a hybrid deterministic-stochastic approach constrained by experimental ensemble data was used for predicting and characterising the input-output behaviour of a single MAPK cycle. This approach revealed that the MEK-ERK cycle in PMA stimulated Jurkat-T cells could be operating in H or ST regimes depending on the strength of the stimulus [21]. A quasi-steady-state approximation Michealis-Menten model [22] employed in the hybrid approach [21] could not explain the observed transition of the regime effected by merely changing the stimulus strength. A question thus arises as to what could be the mechanism that may govern the observed operating regime transition under steady-state conditions.

When Raf/MEK/ERK enzymatic cascades is modelled without an explicit feedback, signal flow is usually described as a one-way communication, that is, going from upstream to downstream of the cascade [23]. However, recently, a new type of signalling called retroactive signalling, which is caused by the presence of a downstream load, has been considered [24–28]. This phenomenon occurs due to the possibility that the futile cycles are coupled with another downstream cascade/substrate. Either or both forms of the protein involved in a futile cycle could be sequestered by another substrate which could be a part of another cascade or simply by a DNA to which one of the forms of the protein is sequestered [24–26, 29, 30]. In the case of Raf/MEK/ERK, the sequestration of the phosphorylated ERK could result in a retroactivity in the cascade. It has been shown experimentally that retroactivity indeed plays a role in the behaviour of MAPK cascades and other signalling pathways [24, 31–35]. Presence of retroactivity in enzymatic cascades has been suggested to predict a more realistic drug-response curve, that is, an input-output behaviour [30].

Inclusion of sequestration effects, which is known to affect the enzymatic futile cycle behaviour [24, 35] may cause a shift in the operating regimes at deterministic level [30]. It is thus likely that incorporating the presence of retroactive signalling might predict the stimulus-strength dependent operating regime transition. In this study, we consider systematically characterising the effect of the presence of substrate or product retroactivity on the operating regimes of MEK/ERK enzymatic cascade. Specifically, we show that strength of the retroactive signalling can modulate the nature of the operating regimes and can permit operating regime transitions.

## 2. Mathematical model of a futile cycle with retroactive signalling

We consider an enzymatic futile cycle with retroactive signalling wherein an enzyme catalysed transition between inactive ($M$) and active ($M_p$) forms of the protein substrate occurs (Fig 1). We assume that both forms of the proteins $M$ and $M_p$ may be sequestered reversibly, respectively by downstream targets $S_1$ and $S_2$ and thereby incorporate retroactivity in the cascade (Fig 1).

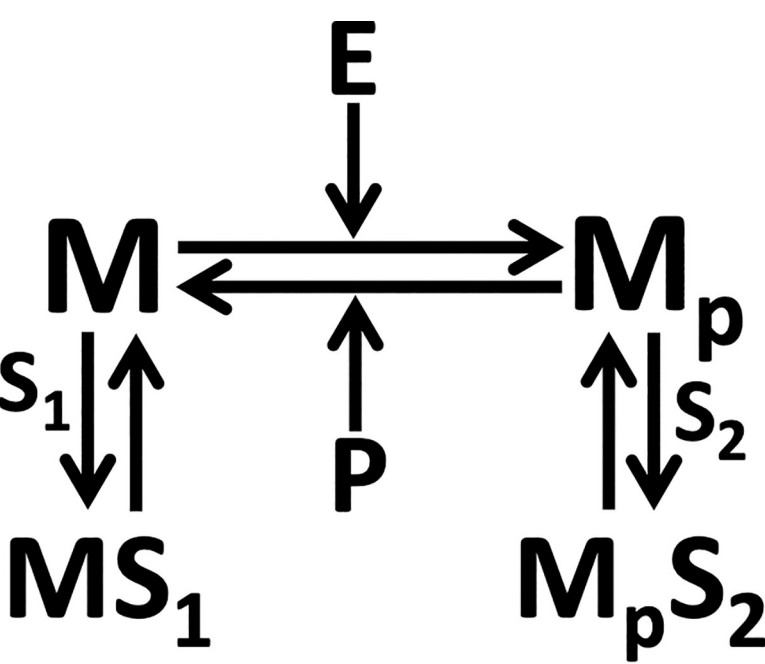

**Fig 1. Enzymatic futile cycle with retroactivity.** M and $M_p$ are the inactive and active forms of the protein substrate. Kinase E and phosphatase P, respectively are the enzymes for the phosphorylation and dephosphorylation biochemical reactions. While $S_1$ and $S_2$ are the downstream targets, respectively of M and $M_p$, $MS_1$ and $M_pS_2$ are the corresponding sequestered complexes.

The biochemical reactions corresponding to the enzymatic cascade in Fig 1 are

$$E + M \rightleftharpoons EM \rightarrow E + M_p \qquad [1]$$

$$P + M_p \rightleftharpoons PM_p \rightarrow P + M \qquad [2]$$

and those capturing the downstream sequestration steps are

$$M + S_1 \rightleftharpoons MS_1 \qquad [3]$$

$$M_p + S_2 \rightleftharpoons M_pS_2 \qquad [4]$$

We assume quasi-steady state approximation (QSSA) for the two intermediate complexes in Eq (1) and (2), and for the two complexes formed by sequestration reactions (Eqs 3 and 4). Upon employing QSSA, the dynamics of dimensionless concentration of $M_p$, $\bar{m} = m_p/m_t$ where $m_t$ is the total protein substrate, dictated by the biochemical reactions in Eqs (1–4) is given by the mathematical kinetic model

$$\frac{d\bar{m}}{dt} = \frac{1}{m_t}(R_p(e_t, \lambda, \bar{m}) - R_d(\alpha, \bar{m})) = \frac{k_f e_t(1 - \bar{m})}{K_1(1 + \lambda) + m_t(1 - \bar{m})} - \frac{k_r p_t \bar{m}}{K_2(1 + \alpha) + m_t \bar{m}} \qquad [5]$$

where, $k_f$ and $k_r$, respectively are the forward and reverse catalytic rate constants, $e_t$ and $p_t$, respectively capture the total concentrations of kinase $E$ and phosphatase $P$. $K_1$ and $K_2$ are the Michaelis-Menten (MM) constants for the forward and backward enzymatic reactions, respectively [22, 36]. $R_p(e_t, \lambda, \bar{m})$ and $R_d(\alpha = 0, \bar{m})$, respectively capture the phosphorylation and dephosphorylation reaction rates. Assuming the equilibrium constants for binding of $M$ and $M_p$ are equal, the retroactivity strengths for sequestration of $M$ and $M_p$, respectively are given

by

$$\lambda = s_1/K_d \text{ and } \alpha = s_2/K_d \tag{6}$$

where $s_1$ and $s_2$ are the concentrations of species $S_1$ and $S_2$, respectively and $K_d$ is the equilibrium constant corresponding to the sequestration reactions. A detailed derivation of Eq (5) from the full model capturing the dynamics of the biochemical reactions (Eqs 1–4) along with the definition of associated MM constants is in S1 Appendix. Note that the effect of retroactivity of either $M$ or $M_p$ or both on the phosphorylation (first term in the right hand side or rhs) and dephosphorylation (second term in rhs) rates in Eq (5) is quantitatively accounted for by *scaling the MM constants $K_1$ and $K_2$ with non-zero (positive) values of $\lambda$ and $\alpha$, respectively.*

Upon setting the left hand side or lhs to zero and solving analytically the resulting quadratic equation, we find the steady-state solution of Eq (5) as

$$\bar{m} = \frac{m_p}{m_t} = \begin{cases} \dfrac{-b + \sqrt{b^2 - 4(k_f e_t/k_r p_t)(1 - k_f e_t/k_r p_t)(K_2(1+\alpha)/m_t)}}{2(k_f e_t/k_r p_t - 1)}, & \dfrac{k_f e_t}{k_r p_t} \neq 1 \\[2em] \dfrac{1}{1 + K_1(1+\lambda)/K_2(1+\alpha)}, & \dfrac{k_f e_t}{k_r p_t} = 1 \end{cases} \tag{7}$$

where, $b = -(k_f e_t/k_r p_t - 1) + (K_2(1+\alpha)/m_t)(k_f e_t/k_r p_t) + K_1(1+\lambda)/m_t$. In S1 Text, we show that this steady-state solution (Eq 7) matches with that of the full model (Eq [AI.1-AI.4] and [AI.5]), for all range of values assigned to the parameters. Dose-response curve $\bar{m}_p$ ($\bar{K}_1$, $\bar{K}_2$, $e_t$) of the futile cycle with (or without) retroactivity is essentially the locus of the relationship between $\bar{m}_p$ and $e_t$, with all other parameters fixed [19, 21]. Note that $\bar{m}_p$ can be drawn using Eq (7) for (a) without retroactivity by setting $\alpha = \lambda = 0$, (b) with retroactivity only in $M$ by setting $\alpha = 0, \lambda > 0$, (c) with retroactivity only in $M_p$ by setting $\alpha > 0, \lambda = 0$ and (d) with retroactivity in both $M$ and $M_p$ by setting $\alpha > 0, \lambda > 0$ [24]. Since introduction of retroactivity tantamount to proportional scaling of the MM constants (Eq 5), for the sake of brevity, we define effective MM constants $\bar{K}_1 = K_1(1+\lambda)$ and $\bar{K}_2 = K_2(1+\alpha)$ which when $\lambda$ or $\alpha$ set to zero will correspond to the case of absence of retroactivity in $M$ or $M_p$, respectively. Dose-response curve $\bar{m}_p$ can be classified into four distinct operating regimes, *viz.*, H, ST, TH and U. Each of these regimes have a representative dose-response curve referred to as nominal profile. These four nominal profiles correspond to the four combinations of the saturated or unsaturated states of the two enzymatic reactions, *viz.*, phosphorylation and dephosphorylation reactions of the futile cycle, as summarized in Table 1. An enzymatic reaction is considered saturated when most of the enzyme is bound to the substrate. The saturated state of the reaction occurs when the corresponding Michaelis-Menten constant is significantly smaller than the substrate concentration. $\bar{K}_1^n, \bar{K}_2^n$, where superscript $n$ = H, ST, TH and U, used for arriving at the four nominal profiles of the futile cycle are in Table 1. As a ready reckoner, we present in Fig 2 the nominal dose-response curves $\bar{m}_p^n$ with $n$ = H, ST, TH, U for the four regimes. Parameters

**Table 1. The nature of the state of the two biochemical reactions corresponding to four operating regimes.**

| Regime | ST | H | TH | U |
|---|---|---|---|---|
| *Phosphorylation reaction* | Saturated | Unsaturated | Unsaturated | Saturated |
| *Dephosphorylation reaction* | Unsaturated | Unsaturated | Saturated | Saturated |
| $\bar{K}_1^n, \bar{K}_2^n$ | 10,10000 | 10000,10000 | 10000,10 | 10,10 |

Michaelis-Menten constants $\bar{K}_1^n, \bar{K}_2^n$ used to arrive at the nominal profiles of the four regimes [18, 21].

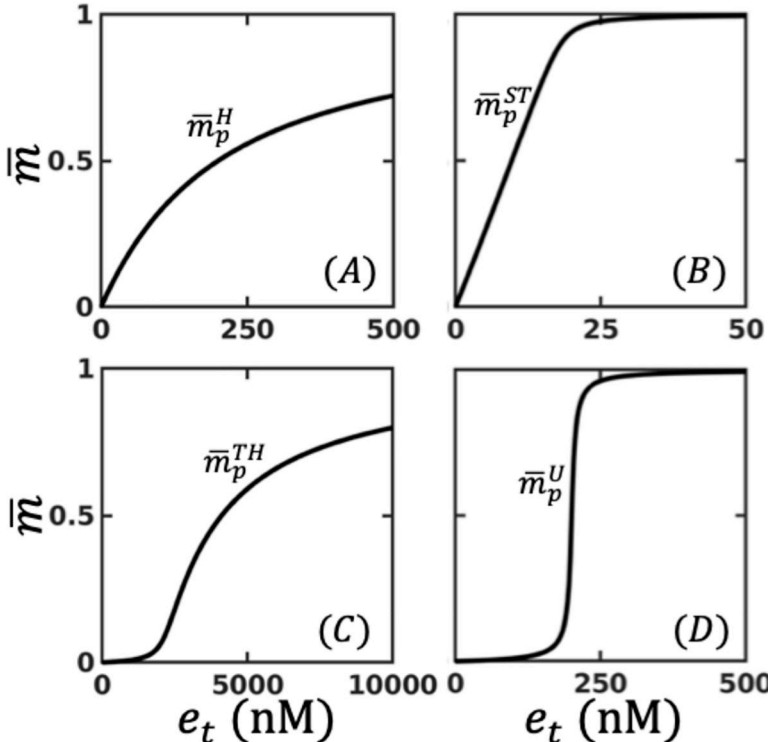

**Fig 2.** Schematic showing the steady-state dose response curve corresponding to the nominal profiles of (A) Hyperbolic ($\bar{m}_p^H$), (B) Signal transducing ($\bar{m}_p^{ST}$), (C) Threshold hyperbolic ($\bar{m}_p^{TH}$), (D) Ultrasensitive ($\bar{m}_p^U$). The conditions employed for simulating these dose-response curves are in Table 1.

besides $\bar{K}_1^n, \bar{K}_2^n$ used for arriving at these curves are $k_r = k_f = 0.01 s^{-1}$, $p_t = 200$nM and $m_t = 1000$nM [18, 21]. Unless otherwise explicitly stated, these parameter values specified are employed for the rest of the study.

While hyperbolic response of the futile cycle is robust to fluctuations and can transmit signals in a broad range of amplitudes [37], signal transducing (ST) regime exhibiting a linear response is amenable for signalling involving graded stimuli. In the threshold-hyperbolic regime, the response of the futile cycle occurs only if the input is above the threshold, after which it increases hyperbolically [38]. Ultrasensitive (U) regime permits amplification of a small signal near the threshold which biological systems take advantage of [39]. As a reference, we employ the nominal profiles corresponding to the case wherein retroactivity is absent.

A dose-response curve is placed in one of the four regimes by contrasting the corresponding $\bar{m}_p(\bar{K}_1, \bar{K}_2, e_t)$ with $\bar{m}_p^n(\bar{K}_1^n, \bar{K}_2^n, e_t) = m_p^n(\bar{K}_1^n, \bar{K}_2^n, e_t)/m_t$, where superscript $n$ = H, ST, TH, U indicates regime-specific nominal profile (Methods). This approach has been suggested by Gomez-Uribe et al. [18] and adopted in several recent studies [21, 40].

## 3. Results

### 3.1 Retroactivity impacts operating regimes

In order to study the effect of retroactivity on the dose-response curve, we adopt the same strategy prescribed by Gomez-Uribe et al. [18] to characterize the operating regimes in the presence of a downstream load on $M$ or $M_P$. We limit the scope of this study to the presence of retroactivity in either $M$ or $M_p$. Systematic characterization reported here, without loss of

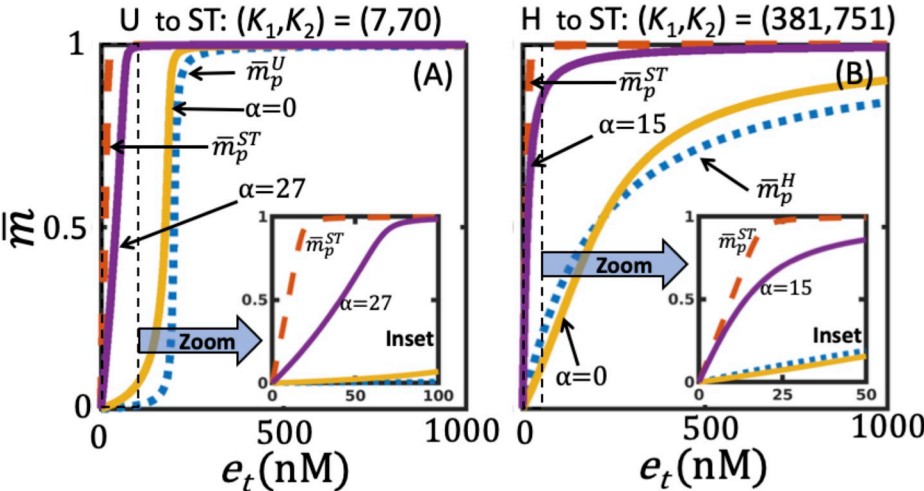

**Fig 3.** Retroactivity in $M_p$ inducing operating regime transition from (A) U at $\bar{K}_1 (\lambda = 0) = 7$, $\bar{K}_2(\alpha = 0) = 70$ to ST at $(\bar{K}_1(0), \bar{K}_2(27)) = (7,1960)$ and (B) H at $(\bar{K}_1(\lambda = 0) = 389$, $\bar{K}_2(\alpha = 0) = 751)$ to ST at $(\bar{K}_1(0), \bar{K}_2(15)) = (389,12016)$. Inset: Zoom in of the dose-response curves. For ease of comparison, the nominal profiles $\bar{m}_p^U$ and $\bar{m}_p^{ST}$ from Fig 2D and 2B, respectively are included in (A). Similarly, $\bar{m}_p^H$ and $\bar{m}_p^{ST}$ from Fig 2A and 2B, respectively are included in (B). Parameters $(\bar{K}_1(0), \bar{K}_2(0))$ used for simulating the nominal profiles are in Table 1.

generalization, can be used for the case where retroactivity may be present in both $M$ and $M_p$, simultaneously.

In order to assess if retroactivity impacts the nature of the operating regime for a certain set of parameters, we consider a dose-response curve in the U regime in the absence of retroactivity ($\alpha = \lambda = 0$), when $(\bar{K}_1(0), \bar{K}_2(0)) = (K_1, K_2) = (7,70)$. Fig 3A shows this dose-response curve (solid yellow) contrasted against the nominal profile for U regime (dashed blue), included from Fig 2D for ease of comparison, used for identifying the regime to which it belongs to. Introduction of retroactivity in $M_p$ with a strength of $\alpha = 27$ (and $\lambda = 0$) resulting in $(\bar{K}_1(0), \bar{K}_2(27) = (7,1960))$ causes shifting of the dose-response curve (solid purple curve in Fig 3A) to the left. $\bar{m}_p$ for case of $\alpha = 27$ belongs to the ST regime indicating the possibility of retroactivity induced transition of operating regimes. (For the sake of comparison, we present $\bar{m}_p^{ST}$ (dashed red curve) from Fig 2B in Fig 3A). We further show that introduction of (a) retroactivity in $M_p$ can induce regime transition from H at $\bar{K}_1(0), \bar{K}_2(0) = (389,751)$ to ST regime at $(\bar{K}_1(0), \bar{K}_2(1.5) = (389,1878)$ (Fig 3B) and (b) retroactivity in $M$ can induce operating regime transition from ST to H (S1 Text). Given that the presence of a downstream load can cause a regime shift, we ask a question as to what are the other possible transitions in the presence of retroactivity. The primary goal of this study is to systematically understand the effect of retroactivity in $M$ or $M_p$ on the operating regimes.

### 3.2 Retroactivity strength dictates nature of regime transition

Retroactivity introduces a scaling for the Michaelis-Menten constants (Eq 5) and thereby affects the steady-state behaviour (Eq 7). As a result, in order to study the effect of retroactivity strength on the operating regimes, it is sufficient to understand how the parameter space of effective Michaelis-Menten constants $\bar{K}_1 = K_1(1+\lambda)$ and $\bar{K}_2 = K_2(1+\alpha)$ is partitioned into different input-output behaviours. Note that replacing $K_1(1+\lambda)$ and $K_2(1+\alpha)$ in Eq (7) respectively with $\bar{K}_1$ and $\bar{K}_2$ makes the retroactivity embedded steady-state solution form similar to that of an isolated enzymatic cascade. Thus, knowledge of the boundaries of the different operating regimes in the planes of $\bar{K}_1$ and $\bar{K}_2$ could be directly used to decipher the effect of

retroactivity on the dose-response curves exhibiting a certain input-output characteristic by varying $\lambda$ or $\alpha$.

Next, we implemented an optimization problem to delineate the parameter space $(\bar{K}_1, \bar{K}_2)$ corresponding to the four distinct operating regimes. For the ease of constructing the map, assuming $\alpha = \lambda = 0$, for an operating regime, after specifying a $\bar{K}_1$ we identified $\bar{K}_2$ by increasing retroactivity strength $\alpha$ such that the candidate dose-response curve $\bar{m}_p^c(\bar{K}_1, \bar{K}_2, e_t)$ satisfied the relative distance criterion

$$d_c(\bar{K}_1, \bar{K}_2, n) = \frac{\|\bar{m}_p^c - \bar{m}_p^n\|}{\max_{\forall c} \|\bar{m}_p^c - \bar{m}_p^n\|} = 0.1 \qquad [8]$$

for all $n$ = H, ST, TH, and U. (Note that superscript $c$ in $\bar{m}_p^c(\bar{K}_1, \bar{K}_2, e_t)$ refers to a candidate.) This criterion is based on the metric suggested by Gomez-Uribe et al. [18] and recently used in Parundekar et al. [21]. In the metric introduced by Gomez-Uribe et al. [Gomez-Uribe et al.], the total substrate concentration is used as scaling. The predictions are therefore a function of the total substrate concentration itself. However, the basis for finding the distance from the nominal curves introduced by Parundekar et al. [21] constitutes scaling using the maximum regime-specific distance from its nominal profile. This metric offers advantages such as scaling being a self-learned parameter, relative distance estimation that is not biased by the system parameters. Next, we briefly describe the procedure adopted for estimating $d_c(\bar{K}_1, \bar{K}_2, n)$.

Every candidate dose-response curve will have four distances, each corresponding to a comparison with four regime-specific nominal profiles (Fig 2). Finding $d_c(\bar{K}_1, \bar{K}_2, n)$ objectively for a dose-response curve requires estimation of $\max_{\forall c} \|\bar{m}_p^c - \bar{m}_p^n\|$ in Eq (8) *a priori*. However, the information about the regime to which a candidate $\bar{m}_p^c$ belongs to is unavailable. In order to address this, we first created a randomly chosen parameter-profile database containing 140000 sets of $(\bar{K}_1, \bar{K}_2)$ sampled using stratified random sampling (Methods) across five orders of magnitude range each tagged to its dose-response curve $\bar{m}_p$. (Note that the maximum possible value that an element in $\bar{m}_p$ can take is 1 [21].) Next, we performed an optimization (Methods) for finding $\bar{K}_2$ that satisfies Eq (8) and its corresponding $\bar{m}_p$. As an example, consider finding the boundaries of H regime by setting $n$ = H in Eq (8). In the five-orders of magnitude range considered, finding $(\bar{K}_1, \bar{K}_2)$ whose corresponding $\bar{m}_p$ satisfied Eq (8) enabled identifying the boundary for the H regime in the planes of effective Michaelis-Menten constants (Fig 4, orange lines). Note that the dashed lines correspond to those $(\bar{K}_1, \bar{K}_2)$ on the boundary sourced directly from the database. We repeated the entire procedure to find the boundaries corresponding to U (blue), ST (yellow), and TH (red) regimes (Fig 4). We note that upper boundary of the ST regime is an exception. While constructing the upper boundary for ST regime, we observed that the dose-response curve is *insensitive* to $\bar{K}_2$ beyond a certain limit after which $\bar{K}_2$ has no effect on $d_c(\bar{K}_1, \bar{K}_2, ST)$. Therefore, for representation purposes, we fixed the upper boundary for ST (yellow) at $d_c(\bar{K}_1, \bar{K}_2, ST) \approx 0.02$ by accordingly modifying Eq 8. Note that as a direct consequence the dose-response curves well beyond the upper boundary of ST will belong to the signal-transducing regime. Metric adopted in Eq 8 by and large separates the regions where these four regimes exist. We note that the underlying model assumptions and the metric used by Gomez-Uribe et al. [18] are different as compared to those considered here. These differences could be attributed to the range for the operating regimes in Fig 4 not being same as those reported in [18].

Since changing retroactivity strength can independently modulate the Michaelis-Menten constants, manipulating $\bar{K}_1$ or $\bar{K}_2$ or both could cause a shift in the characteristic input-output behaviour. Specifically, by increasing the strength of the load in $M$ causing proportional change in $\bar{K}_1$ while keeping $\bar{K}_2$ constant, a dose-response curve in U or ST, respectively can

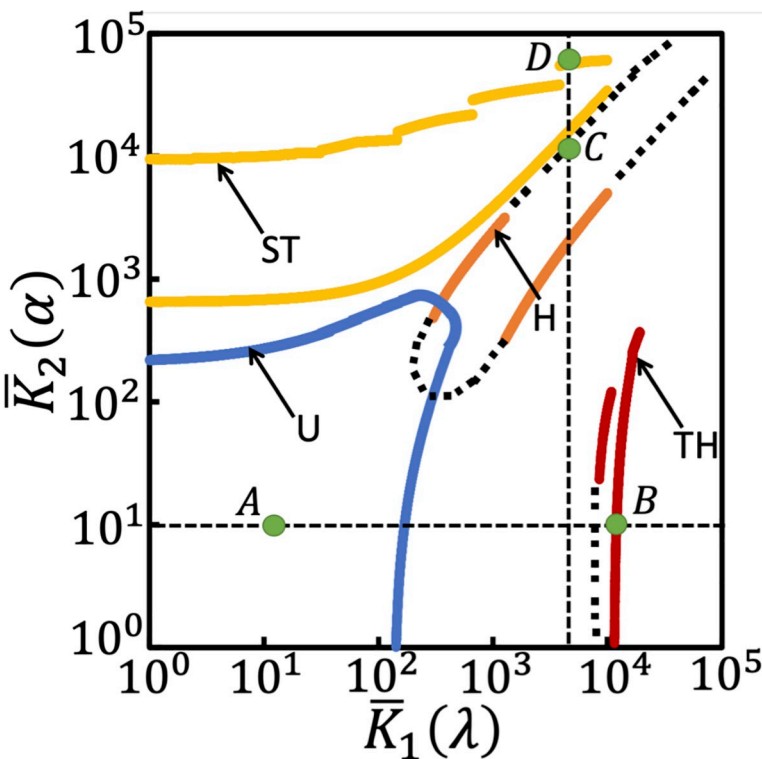

**Fig 4. Boundaries of the four operating regimes hyperbolic (H, blue), signal transducing (ST, green), threshold-hyperbolic (TH, red), and ultrasensitive (U, yellow) in the planes of $\bar{K}_1$ and $\bar{K}_2$ for $\lambda = \alpha = 0$.** All boundaries of each of the regimes except the upper boundary of ST satisfy the relative distance criterion in Eq (8). For the case of upper boundary of ST the rhs of Eq (11) was set to 0.02. ($\bar{K}_1$, $\bar{K}_2$) on the dotted lines extending the solid line boundaries were sourced directly from the database. While dose-response curves corresponding to parameter sets at green dots A and B were used as example of transition from U to TH regime, those curves at C and D of transition from H to ST.

shift to TH or H. For example, $\bar{m}_p(10,10)$ in U regime (Fig 4, point A) would shift to TH (Fig 4, point B) upon increasing $\bar{K}_1$ to 10000. Similarly, while maintaining $\bar{K}_1$ constant, an increase in the retroactivity strength in $M_p$ leading to proportional change in $\bar{K}_2$ could lead to four other possible regime transitions, *viz.*, U to ST, TH to H or ST, and H to ST. $\bar{m}_p$ (6000,6000) in H regime (Fig 4, point C) transitions into ST regime (Fig 4, point D) when $\bar{K}_2$ is scaled to 60000. For a given source profile specified by a certain ($\bar{K}_1$, $\bar{K}_2$) with no retroactivity either in $M$ or $M_p$, while maintaining $\bar{K}_1$ or $\bar{K}_2$ constant, the minimum load $\lambda_{min}$ or $\alpha_{min}$, respectively required for inducing a regime transition is sensitive to the chosen $\bar{K}_1(0)$ or $\bar{K}_2(0)$ (S1 Text). This sensitivity analysis showed the minimum load needed for any regime transition to occur is 0.3. This minimum corresponds to transition of ST at ($\bar{K}_1(\lambda = 0), \bar{K}_2(\alpha = 0)$) = (8205,28160) to H regime due to retroactivity in $M$ with $\lambda_{min}$ being 0.3. $\lambda_{min} = 0.3$ translates to ($ms_1/(m_t - m_p)$) = $\lambda_{min}/(1 + \lambda_{min})$ = 0.23 indicating that 23% of the unphosphorylated substrate sequestered by downstream target is needed for inducing this transition.

### 3.3 Saturation level of the two enzymatic reactions governs the retroactivity induced regime transition

Since the dose-response curve $\bar{m}_p$ explicitly depends on $\bar{K}_1(\lambda)$ and $\bar{K}_2$ ($\alpha$) (Eq 7), understanding how load strength $\lambda$ or $\alpha$ influences the input-output behaviour may offer useful insights into what causes retroactivity driven operating regime transition. In order to assess the

regime-specific impact of retroactivity on the input-output behaviour, we systematically ana-lyse the dose-response curves and the associated sensitivity with respect to retroactivity strengths $\lambda$ and $\alpha$.

The sensitivity of $\bar{m}$ with respect to retroactivity strength $\lambda$ and $\alpha$, respectively are quantita-tively captured by

$$\frac{d\bar{m}}{d\lambda} = \left(\frac{d\bar{m}}{d\bar{K}_1}\right)\left(\frac{d\bar{K}_1}{d\lambda}\right) = \left(\frac{d\bar{m}}{d\bar{K}_1}\right)\bar{K}_1(0) = \left(\frac{d\bar{m}}{d\bar{K}_1}\right)K_1 \qquad [9]$$

and

$$\frac{d\bar{m}}{d\alpha} = \left(\frac{d\bar{m}}{d\bar{K}_2}\right)\left(\frac{d\bar{K}_2}{d\alpha}\right) = \left(\frac{d\bar{m}}{d\bar{K}_2}\right)\bar{K}_2(0) = \left(\frac{d\bar{m}}{d\bar{K}_2}\right)K_2 \qquad [10]$$

for a finite (non-zero) downstream load. Detailed expressions of these are in S2 Appendix. Eqs 9 and 10 show that the presence of retroactivity in $M$ or $M_p$ introduces a constant scaling of $\bar{K}_1(0) = K_1$ or $\bar{K}_2(0) = K_2$, respectively to the sensitivity with respect to the corresponding Michaelis-Menten constant. In the sub-sections below, we present the sensitivity effects due to modulation of retroactivity corresponding to either $M$ or $M_p$ for these five transitions and distil out the underlying causal mechanism. For the case of substrate or product retroactivity modu-lation, we first fix $(\bar{K}_1(0) = K_1, \bar{K}_2(0) = K_2)$ in a certain regime with no retroactivity and then increasing $\lambda$ or $\alpha$, respectively and track the ensuing regime transition.

**3.3.1 U to TH transition due to retroactivity in M.** The dose-response curves obtained by starting from $U$ regime for $(\bar{K}_1(\lambda = 0) = K_1, \bar{K}_2(\alpha = 0) = K_2) = (9\text{nM},9\text{nM})$ with $d_c(9\text{nM},9\text{nM}, U) = 0.013$ and transitioning into $TH$ regime by changing $\lambda$ is shown in Fig 5A. Note that while $\alpha = 0$, increasing $\lambda$ leads to a proportional scaling of $\bar{K}_1(\lambda)$. Introduction of ret-roactivity causes changes to the extent of ultrasensitive nature of the dose-response curves. This extent of ultrasensitive nature in the presence of retroactivity can be quantified via the half-maximal response given by

$$\mathcal{S}_{50} = \frac{k_r p_t}{k_f EC50} = \frac{\bar{K}_2(\alpha) + 0.5}{\bar{K}_1(\lambda) + 0.5} = \frac{K_2(1+\alpha) + 0.5}{K_1(1+\lambda) + 0.5} \qquad [11]$$

which uniquely specifies the dose-response curve's EC50, that is, $e_t$ at which $\bar{m} = 0.5$ [24]. Note that when $\lambda = \alpha = 0$, Eq (11) reduces to the response defined in Goldbeter and Koshland [19]. As the dose-response curve transits from U to TH, the EC50 increases from 200 to 4400 for the range of $\lambda$ considered (S1.3 Fig in S1 Text). Note that EC50 increases linearly with the retroac-tivity strength $\lambda$ (Eq 11). Moreover, Fig 5A also reveals that an increase in load shifts the dose-response curve by simultaneously enlarging the curve's base resulting in a threshold and also the curvature eventually leading to a TH input-output behavior. Next, we elucidate what causes the observed U to TH transition.

In Fig 5B, we show the modulation of sensitivity (Eq 9) by $\lambda$ and $e_t$. An increase in $\lambda$ in dose-response curve $\bar{m}_p(\bar{K}_1(\lambda), \bar{K}_2(0), e_t)$ leads to a decreased negative sensitivity. This shift in peak is correlated to the corresponding increase in the EC50 (S1.3 Fig in S1 Text), as has also been reported in Ventura et al. [24]. This behaviour is dictated by the steady-state levels of $M_p$ (Eq 7), which is a balance between the phosphorylation and dephosphorylation rate terms in the rhs of Eq 6 for a given $\lambda$ and $e_t$. Insights into the effect of $\lambda$ on $\bar{m}$ can be deciphered from the nature of relative variation of these two rates, which we discuss next.

In Fig 5C, we present the rate-balance plot consisting of the rate curves of the phosphoryla-tion reaction $R_p(e_t = 1000 \text{ nM}, \lambda, \bar{m})$ for different $\lambda$ and of the dephosphorylation reaction $R_d(\alpha = 0, \bar{m})$. Note that $R_p$ and $R_d$ are the rates of the two enzymatic reactions defined in Eq

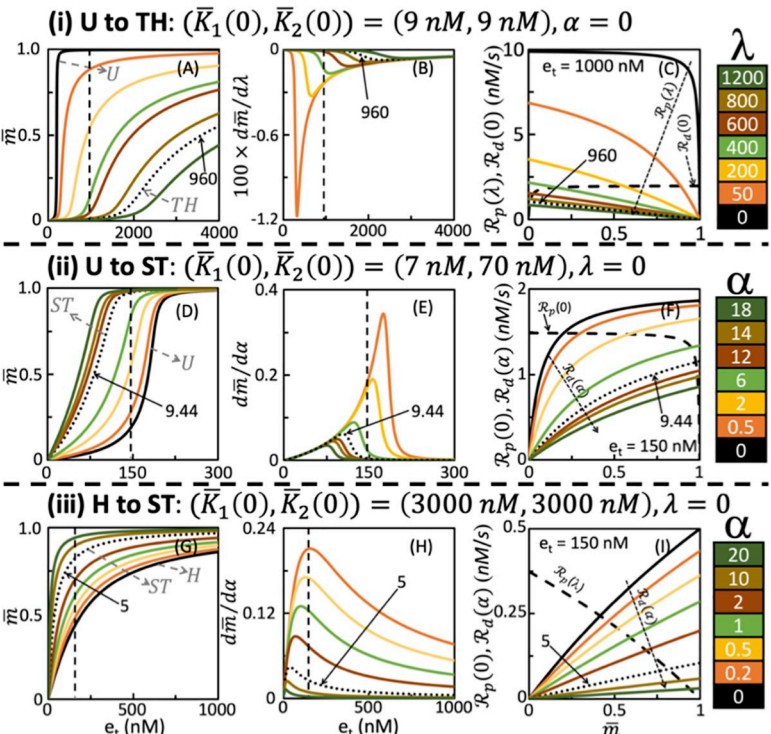

**Fig 5.** Effect of retroactivity strength on the operating regimes and the associated sensitivity for (i) U to TH, (ii) U to ST, and (iii) H to ST transitions. While panel (i) corresponds to effects due to load on M quantified by $\lambda$, panel (ii) and (iii) captures those due to load on $M_p$ quantified by $\alpha$. Dependence of dose-response curves on the load corresponding to (i), (ii) and (iii) are in (A), (D) and (G), respectively. Sensitivity of steady-state level for different retroactivity strengths for (i), (ii), and (iii) are in (B), (E) and (H), respectively. Sensitivity curves in (B) was estimated using Eq (9), Eq (10) was employed for those in (E) and (H). While rate-balance plot showing the effect of retroactivity strength on modulation of steady-state levels corresponding to (i) at $e_t$ = 1000nM is in (C), that for (ii) and (iii) at $e_t$ = 150nM are in (F) and (I), respectively. Colorbar in each of the panels display the retroactivity strengths. Dotted line in (A), (B) and (C) in panel (i) corresponds to the dose-response, sensitivity and $R_d(\lambda)$ curves, respectively at the transition where $\lambda$ = 960. Dotted line in panels (ii) and (iii) captured these curves at the corresponding transition where $\alpha$ = 9.44 and $\alpha$ = 5, respectively.

(5). The nature of an enzymatic reaction being saturated, that is, all enzymes bound to its substrate, is specified by the range of $\bar{m}$ for which the corresponding rate does not change significantly. Therefore a sufficient proportional increase in $\bar{K}_1(\lambda)$ due to $\lambda$ can lead to $R_p$ exhibiting a linear dependence on $\bar{m}$ in a certain range. The nature of the phosphorylation reaction is unsaturated in this range of $\bar{m}$. In the U regime, both $R_p(1000 \text{ nM}, \lambda = 0, \bar{m})$ (Fig 5C, black) and $R_d(0, \bar{m})$ (Fig 5C, dashed) curves are predominantly saturated. For the chosen $e_t$, at $\lambda = 0$, the intersection occurs in the region where $R_p$ is *not* saturated and $R_d$ is saturated, leading to $\bar{m} \approx 1$. Increasing $\lambda$ forces the phosphorylation reaction ($R_p$ curve) to gradually become predominantly unsaturated (Fig 5C). The extent of this unsaturation introduced underlies the shift in the intersection point of the rate curves in the direction of decreasing $\bar{m}$. Thus, increasing $\lambda$ causes significant decrease in the steady-state levels $\bar{m}$ (Fig 5C). This decrease explains the gradual change in the steady-state levels at $e_t$ = 1000 nM in the different dose-response curves in Fig 5A. Moreover, this decrease results in a significant change in the sensitivity (Fig 5B). At $\lambda$ = 960, due to sufficient levels of unsaturation, the operating regime transits into the TH regime, which is characterized by the phosphorylation and dephosphorylation reactions, respectively being unsaturated and saturated. At the transition, the relative distance from the TH nominal profile $\bar{m}_p^{TH}$ is $d_c(\bar{K}_1(960) = 8649, \bar{K}_2 = 9, \text{TH}) = 0.0833$.

**3.3.2 U to ST transition due to retroactivity in $M_p$.** For $(\bar{K}_1(\lambda = 0) = K_1, \bar{K}_2(\alpha = 0) = K_2)$ = (7nM,70nM), the effect of dose-response curves on the retroactivity strength $\alpha$ is in Fig 5D. The dose-response curve when $\alpha = 0$ (Fig 5D, black) with a $d_t$(7nM,70nM, U) = 0.049 and EC50 of 178 nM, at $\alpha = 9.44$ shifted to the ST regime with a $d_t(\bar{K}_1(0) = 7, \bar{K}_2(9.44) \approx 731, ST) = 0.0965$, with the EC50 being 82 nM (S1.3 Fig in S1 Text). We next discuss what causes this regime transition.

Fig 5E shows that an increase in the retroactivity strength $\alpha$ in dose-response curve $\bar{m}_p$ ($\bar{K}_1$(0), $\bar{K}_2(\alpha)$, $e_t$) while maintaining $\lambda = 0$ causes reduction in the (positive) sensitivity. The rate-balance analysis at $e_t$ = 150nM shows that when $\alpha = 0$, the intersection of the two rate-curves occurs in the region where the phosphorylation reaction is near saturation (Fig 5F). Note that the $R_d$ curve is predominantly saturated when $\alpha = 0$. An increase in $\alpha$, that is, $\bar{K}_2(\alpha)$ shifts the nature of $R_d$ curve to predominantly unsaturated. This shift causes the intersection, that is, steady-state level, to increase from 0.2 at $\alpha = 0$ to 0.99 at $\alpha = 18$. Therefore, increasing the load leads to an increase in the steady-state level depending on the extent of the unsaturation evidenced by the dephosphorylation reaction. This shift in the steady-state level forces the dose-response curve to move into the ST operating regime.

**3.3.3 H to ST transition due to retroactivity Mp.** For $(\bar{K}_1(\lambda = 0) = K_1, \bar{K}_2(\alpha = 0) = K_2)$ = (3000nM,3000nM), the effect of dose-response curves on the retroactivity strength $\alpha$ is in Fig 5G. At $\alpha = 0$, the dose-response curve belonged to the H regime with a $d_c$(3000nM,3000nM, H) = 0.014. Upon increasing $\alpha$ to 5, dose-response curve transitions to ST operating regime with $d_c(\bar{K}_1(0) = 3000, \bar{K}_2(5) \approx 18000, ST) = 0.08$. EC50 for the dose-response curves changes from 200nM to ~38nM (S1.3 Fig in S1 Text). Increase in the load causes a decrease in the sensitivity. The rate-balance plot for $e_t$ = 150nM shows that both phosphorylation and dephosphorylation reactions are predominantly unsaturated in the H regime for $\alpha = 0$. Upon increasing the load, while the dephosphorylation reaction continues to remain unsaturated, the rate curve shows a slowed-down response to increase in $\bar{m}$, that is, reduction of the slope of the $R_d$ curve. This reduction causes a shift in the intersection of the rate-curves to a larger substrate concentration. For e.g., for $e_t$ = 150nM, the steady-state level at $\alpha = 0$ and 20 are 0.42 and 0.94, respectively. Thus, the dose-response curve transitioning from the H to ST is essentially caused by this reduction in the slope of the $R_d$ curve with increase in the retroactivity strength in $M_p$.

**3.3.4 ST to H and TH to H transitions due to retroactivity M and Mp.** In Fig 6, we show the dose-response curves capturing the regime transition from ST to H and TH to H driven by load in $M$ and $M_p$, respectively. For the case of ST to H transition (Fig 6A), at $(\bar{K}_1(\lambda = 0) = K_1, \bar{K}_2(\alpha = 0) = K_2)$ = (9 nM,9000nM), the dose-response curve has an EC50 of ~11 with a $d_c$(9,9000,ST) = 0.003. With an increase in the retroactivity strength $\lambda$ in $M$, the EC50 increases and at $\lambda = 400$, the dose-response curves achieved corresponds to the H regime with a $d_c(\bar{K}_1(400) = 3609, \bar{K}_2 = 9000, TH) = 0.092$ and EC50 of 86.5 (S1.3 Fig in S1 Text). Sensitivity analysis and rate-balance plot at $e_t$ = 12nM show that while dephosphorylation reaction is unsaturated, an increase in $\lambda$ the phosphorylation reaction transitions from predominantly saturated to unsaturated state and thereby, driving the regime transition (S1.4 Fig in S1 Text).

The dose-response curve at $(\bar{K}_1(\lambda = 0) = K_1, \bar{K}_2(\alpha = 0) = K_2)$ = (9000 nM,9 nM) belonging to the TH regime with $d_c$(9000,9,TH) = 0.05 transitions into H regime with increase in load $\alpha$ from 0 to 578 (Fig 6B). The $d_c(\bar{K}_1(0) = 9000, \bar{K}_2(578) = 5211, H) = 0.07$. Sensitivity and rate-balance plots at $e_t$ = 3000 nM suggests that regime transition is caused by the phosphorylation reaction being unsaturated and the dephosphorylation reaction shifting from being predominantly saturated at $\alpha = 0$ to primarily unsaturated with increasing $\alpha$ (S1.5 Fig in S1 Text).

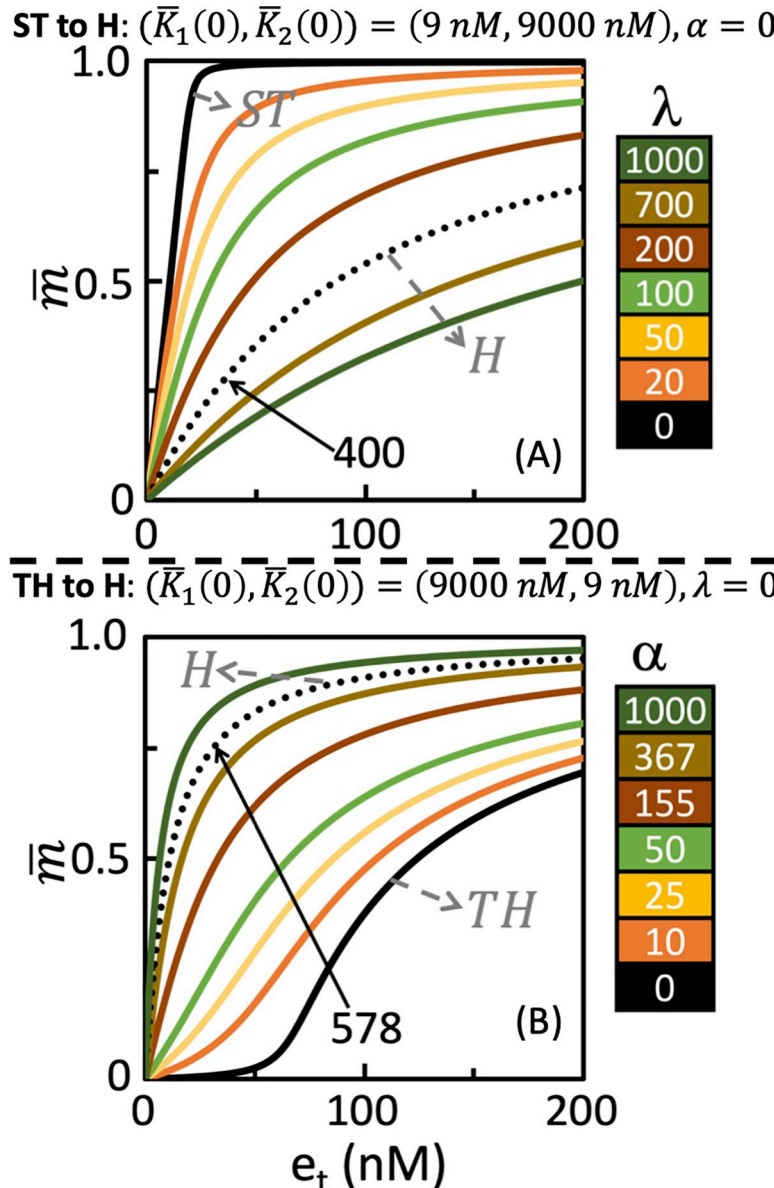

**Fig 6.** Dose-response curves capturing the retroactivity driven transition of operating regimes from (A) ST to H and (B) TH to H. Colorbar displays the retroactivity strength corresponding to the dose-response curves. Dotted lines in (A) and (B), respectively correspond to the retroactivity strength $\lambda = 400$ and $\alpha = 578$ at which the regime transition occurs.

## 4. Discussion and conclusion

Input-output behaviour of an activated enzymatic futile cycle has been studied extensively due to its ability to orchestrate cell fate in direct and indirect context-dependent manners [2, 14, 21]. Micheal-Menten constants (MM) dictated saturated/unsaturated state of the two enzymatic reactions facilitates placing steady-state dose-response curves of a futile cycle into Signal transducing (ST), Hyperbolic (H), Threshold hyperbolic (TH) and Ultrasensitive (U) operating regimes (Fig 2) [18]. The unphosphorylated ($M$) and phosphorylated ($M_p$) forms of the protein substrate involved in the futile cycle can be sequestered by respective downstream

targets. The sequestration dictated load or retroactivity on the upstream protein levels introduces a two-way signal flow permitting modulation of the steady-state behaviour [24, 26]. In this study, we systematically show that the presence of retroactivity in $M$ or $M_p$ can shift the input-output behaviour from one operating regime to another by modulating the level of saturation or unsaturation of the enzymatic reactions. In particular, we demonstrate five possible transitions: (a) U to TH and ST to H caused by retroactivity in $M$ and (b) U to ST, TH to H, and H to ST by that in $M_p$.

Using a quasi-steady state approximated model of the futile cycle with retroactivity, we systematically identified the MM constants' range that permit four distinct operating regimes in the presence of retroactive signalling (Fig 4). Surprisingly, the minimum retroactivity strength needed for inducing any transition is 0.3 which translates to 23% of the substrate bound to its target. For this minimum retroactivity strength of 0.3, dose-response curve at $(\bar{K}_1(\lambda = 0), \bar{K}_2(\alpha = 0)) = (8205, 28160)$ belonging to ST regime transitions into H regime. Several downstream targets that could sequester proteins in MAPK cascades have been reported [35]. Thus, while analysing such a behaviour in a futile cycle using experimental data, ignoring the hidden retroactive signalling effect, however small, could lead to an incorrect prediction of the underlying operating regime.

While in this study we only considered increasing the retroactivity strength to trigger a regime transition, in principle, if downstream sequestrations were already present, its strength can be decreased too. Decreasing the retroactivity strength can predict five other transitions that are essentially the reverse of those analysed in this study. Further, simultaneous increase (decrease) of the retroactivity strengths in $M$ and $M_p$ can lead to a transition from U to H (H to U). Thus, the operating regime boundaries reported in Fig 4 permits prediction of all 12 possible regime transitions. We further note that the U and H regimes have a slight overlap in the $(\bar{K}_1, \bar{K}_2)$ space. Our study indicates that recent experimental observations that a stimulus-strength dependent shift in the operating regimes is possible in a single MAPK cascade if the stimulus concentration change could cause retroactivity induced regime transition [21].

Using sensitivity and rate-balance analysis, we demonstrated that modulation of the saturation or unsaturation levels of the two enzymatic reactions by changing the retroactivity strength is the fundamental reason for the operating regime transition. In particular, we show that increase in retroactivity (a) in $M$ leads to increasing unsaturation in the phosphorylation reaction and (b) in $M_p$ makes dephosphorylation reaction more unsaturated (Fig 5, S1.4 and S1.5 Figs in S1 Text). This is due to the fact that the steady-state level of $M_p$, the active form, is sensitive to changes in the retroactivity strength. While increasing the strength of retroactivity in $M$ causes a decrease in the (negative) sensitivity of the steady-state level, that in $M_p$ leads to marked reduction in the (positive) sensitivity. This sensitivity to retroactive signalling can be capitalized upon to modulate the nature of response of the futile cycle. Synthetic biology tools are becoming available for tweaking the binding sites of targets to which the protein substrate, active/inactive forms may bind and thereby enabling control of the extent of sequestration [41, 42]. The nature of sensitivity effect that retroactive signalling bestows on the steady-state levels demonstrated in this study can be of immense value for precise engineering of a cell to control and modulate the input-output behaviour.

## 5. Methods

### 5.1 Regime identification

The regime that a dose-response curve $\bar{m}_p$ belongs to is identified by contrasting it with the four nominal profiles $\bar{m}_p^n(\bar{K}_1^n, \bar{K}_2^n, e_t)$ where superscript $n$ = H, ST, TH, U. The dose-response

curve $\bar{m}_p$ is ascribed to a certain regime H, ST, TH, or U if the relative distance between $\bar{m}_p$ and the corresponding nominal profile is within 10%.

### 5.2 Stratified random sampling

The two stratification cut-off points were chosen. While choosing these cut-off points, (a) 60000 samples were chosen in the ($0 < K_1 < 1600$, $0 < K_2 < 1600$) and (b) 10000 samples each in the range ($0 < K_1 < 50$, $0 < K_2 < 10000$) and ($0 < K_1 < 10000$, $0 < K_2 < 50$) were chosen [21]. In both these cases uniform distribution was used for sampling. Samples corresponding to either one or both reactions being saturated was at least 10% more than that for the case where both reactions were unsaturated.

### 5.3 Optimization for finding operating regime boundaries

The boundary for a specific regime was obtained by seeking $\bar{K}_2$ that satisfied the objective function in Eq (8) solved using nonlinear optimization function "*fmincon*" implemented in Matlab® [43]. A tolerance of $1e^{-6}$ was set as convergence criteria to the optimization problem. Optimizer convergence was sluggish in the presence of steep gradients and in these cases, the ($\bar{K}_1$, $\bar{K}_2$) samples from the database that satisfied Eq (8) was used.

## Supporting information

**S1 Text.** S1.1- Steady-state solution of the full model. S1.2- Dose-response curve transition from Signal-Transducing to Hyperbolic regime induced by substrate retroactivity. S1.3- Minimum retroactivity strength required to transition from one regime to another. S1.4- Effect of retroactivity strength on EC50 during different regime transitions. S1.5-Sensitivity and rate balance analysis to investigate retroactivity induced ST to H and TH to H regime transitions. (PDF)

**S1 Graphical abstract.**
(TIFF)

**S1 Appendix. Derivation of the mathematical kinetic model.**
(DOCX)

**S2 Appendix. Sensitivity of steady-state level to retroactivity strength.**
(DOCX)

## Acknowledgments

We acknowledge the DBT-Pan IIT Center for Bioenergy, IIT Bombay for an access to the high-performance computing facility.

## Author Contributions

**Conceptualization:** Akshay Parundekar, Ganesh A. Viswanathan.

**Data curation:** Akshay Parundekar, Ganesh A. Viswanathan.

**Formal analysis:** Akshay Parundekar, Ganesh A. Viswanathan.

**Funding acquisition:** Ganesh A. Viswanathan.

**Investigation:** Akshay Parundekar, Ganesh A. Viswanathan.

**Methodology:** Akshay Parundekar, Ganesh A. Viswanathan.

**Project administration:** Ganesh A. Viswanathan.

**Resources:** Akshay Parundekar, Ganesh A. Viswanathan.

**Software:** Akshay Parundekar.

**Supervision:** Ganesh A. Viswanathan.

**Validation:** Akshay Parundekar, Ganesh A. Viswanathan.

**Visualization:** Akshay Parundekar.

**Writing – original draft:** Ganesh A. Viswanathan.

**Writing – review & editing:** Akshay Parundekar, Ganesh A. Viswanathan.

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
