## [Decision Letter · Decision Letter 0]

9 Feb 2021

PONE-D-20-35461

Retroactivity induced operating regime transition in a phosphorylation-dephosphorylation reaction cycle

PLOS ONE

Dear Dr. Viswanathan,

Thank you for submitting your manuscript to PLOS ONE. After careful consideration, we feel that it has merit but does not fully meet PLOS ONE’s publication criteria as it currently stands. Therefore, we invite you to submit a revised version of the manuscript that addresses the points raised during the review process.

We look forward to receiving your revised manuscript.

Kind regards,

Christopher V. Rao

Academic Editor

PLOS ONE

Journal Requirements:

2.Thank you for stating the following in the Acknowledgments Section of your manuscript:

"We thank Department of Science and Technology, 390 Government of India for funding this study."

Reviewers' comments:

Reviewer's Responses to Questions

**Comments to the Author**

1. Is the manuscript technically sound, and do the data support the conclusions?

Reviewer #1: Partly

Reviewer #2: Partly

2. Has the statistical analysis been performed appropriately and rigorously? 

Reviewer #1: N/A

Reviewer #2: N/A

3. Have the authors made all data underlying the findings in their manuscript fully available?

Reviewer #1: Yes

Reviewer #2: Yes

4. Is the manuscript presented in an intelligible fashion and written in standard English?

Reviewer #1: No

Reviewer #2: Yes

5. Review Comments to the Author

Reviewer #1: The study presented by Parundekar and Viswanathan investigates how sequestering the substrate or product of a phosphorylation/dephosphorylation cycle can shape the steady-state dose-response curve of the cycle, where the “dose” or input of the system corresponds to the concentration of the kinase catalyzing the phosphorylation. As a starting point, the authors derived an analytical expression for the steady-state phosphorylation of a simple model of phosphorylation/dephosphorylation including the sequestration of the substrate and/or product. The model output is studied by applying a classification scheme introduced by Gomez-Uribe et al. which categorizes dose-response curves into one of four characteristic operating-regimes such as hyperbolic or ultrasensitive. The main finding of the study is that protein sequestration can alter the shape of the dose-response curve and that by modifying the strength of sequestration, a cell could cause the dose-response curve of the phosphorylation/dephosphorylation cycle to shift from one operating regime into a different one.

Phosphorylation/dephosphorylation cycles are important and ubiqituous signaling motifs and the presented findings are interesting. Although the chosen approach is in principle rigorous, the current implementation seems to be flawed at several points. Please see the list of major and minor concerns below, many of which should be easy to address.

Major issues:

(1) Most importantly: the derivation of equation [5] presented by the authors is, I believe, incorrect and some of the definitions for composite quantities provided by the authors are not consistent with equation [5]. Please refer to the attached document for further details and see whether you agree with my analysis. Generally, I think the derivation presented in appendix 1 would benefit from showing the full mass-action reaction scheme and smaller steps in the derivation

(2) The authors introduce the four operating regimes as defined by Gomez-Uribe with almost no explanation. Since many biologists or biochemists are not even familiar with the term ultrasensitivity, I strongly suggest to add a bit more detail here: How are these regimes defined? How do their dose-response curves look like? (A small figure similar to Fig 2 in Gomez-Uribe et al. 2007 would be helpful.) What are their properties, how does this translate into different signaling functions and why is this biologically important?

(3) It is not clear to me how the regime identification procedure based on relative distance/error can avoid misclassification. For instance, would a TH curve not be misclassified as ultrasensitive if the threshold coincides with the EC50 of the nominal US profile? (see attached picture: TH* is closer to US than to TH). Also, why is the area of the regimes (especially TH) in the K1 vs K2 plot so different from Gomez-Uribe et al. 2007?

(4) Since a regime transition due to sequestration has already been described before (Ventura et al.), the manuscript could benefit from extending the scope a little. Given that Gomez-Uribe et al. 2007 found the different regimes to influence low-pass filtering properties, I suspect studying the influence of k_on or k_off of the sequestration reactions on cycle dynamics and low-pass filtering could offer interesting insights which may increase the impact of the study (e.g. simple numerical analysis with the mass-action model).

Minor issues:

(1) Abstract, p2. Line 35: I don’t think “sequestration strength of 0.3” is a helpful description for most readers. A qualitative description would be more helpful.

(2) The expression “PdPC” seems a bit cumbersome to me. Since the studied motif not only applies to phosphorylation/dephosphorylation cycles but also to other PTMs or to GTPase cycles, I find it more appropriate to use a more common and general expression such as activation/inactivation cycle, PTM cycle, futile cycle …

(3) I don’t understand how increasing the input dose (e.g. kinase concentration) of a cycle can itself induce a regime shift? (p.4, line 77)

(4) p.5, line 118-119: the authors speak of a QSSA for MS1 and MS2, yet I see no use of MS1 or MS2 QSSA in the whole paper

(5) Where does equation [7] come from? From the cited references or from the authors’ study solved manually/by computer algebra system?

(6) Figure 2: The plot of the nominal regime ST almost overlaps with the Y axis and upper border of the figure. The characteristic feature of the ST regime (linear increase until saturation) cannot be identified here! Better rescale the axis or use log(X) axis.

(7) The rate balance figures (figure 4 and supplementary figures) are quite crowded and it would help to use color gradient lines (whose numerical values are given in a legend) instead of all the arrows and values. For figure 4 in particular, it would be very helpful to also show 3D dose-response plots visualising the regime transitions (Z-axis = m, X-axis = e_t, Y-axis = α or λ). These are often more intuitive to understand than rate balance plots.

(8) Sections 3.3.1 and 3.3.2 are largely descriptive of the figures and could be shortened by focussing more on the overall effect of changing α or λ at a nominal operating point and by moving parameter values to the figure legends.

Reviewer #2: In this paper, the authors investigate the operating regimes of a signaling cycle consisting of a protein that can be in an inactive or active form: the protein is activated and deactivated by two enzymatic species, a kinase and phosphatase, respectively. The main novelty introduced in this paper is the presence of a downstream load on the protein (on both the forms) determining retroactivity. The authors characterize the effect of retroactivity on the input-output relationship (i.e. operating regime) of the cycle, where the input is the kinase concentration and the output the active form of the protein (normalized with respect to its total concentration). Moreover, they find that increasing retroactivity strength can trigger five possible regime transitions: for four possible transitions, they show that the modulation of the saturation levels of the enzymatic reactions by increasing retroactivity is the main reason for the operating regime transitions.

The paper is interesting. I have the following comments/questions.

The quasi-steady-state approximation (QSSA) is employed to study the system defined by biochemical reactions [1-4] and get Eq. [5]. However, the full ODE model should also be implemented and simulated: the corresponding steady-state results should be compared with those obtained by exploiting QSSA in order to verify and validate this approximation.

About Fig. 2, it would be better to use inserts in panels A and B showing a zoom-in of the active form of protein vs the kinase concentration (e_t) on lower e_t values.

Moreover, be sure to use a small step size for lower e_t values in order to not miss any significant behavior and confirm the obtained results.

From this study, it seems to emerge that the ultrasensitive threshold (i.e. EC50) can be modulated by retroactivity. Could the authors give more details, in particular, by quantifying the modulation?

Could the authors give more details about the criterion defined by Eq. 8?

For each figure, the caption should be more exhaustive.

I would recommend to the authors to improve understanding/readability of the manuscript to provide more details in the figure captions.

Could the authors explain better the results reported in Section 3.3 and shown in Fig.4? Is the dotted line reported in each panel representing the transition from a regime to another (from U to TH for panels A and B and from U to ST for panels C and D)? Could the authors explain better in the text how retroactivity strength modulates the saturation levels of the enzymatic reactions?

Instead of summarizing the results obtained for the other transitions at the end of Section 3.3.2, it would be better to add another subsection and provide more details of the results reported in the supplementary text.

Minor comments

In Section 2, when the enzyme concentrations e_t and p_t are defined, it should be specified that these concentrations represent the total concentrations of E and P, respectively, in the unbounded and bounded forms.

Please check punctuation, as line 210 (it is missing a full stop).

Define the abbreviations (as rhs) at first occurrence.

Line 273, it should be et=3000 nM in Rp.

Line 275, it should be lambda=0 in Rp.

Line 278, falls instead of fall.

6. PLOS authors have the option to publish the peer review history of their article (what does this mean?). If published, this will include your full peer review and any attached files.

Reviewer #1: **Yes: **Daniel Koch

Reviewer #2: No

---

## [Author Response · Author response to Decision Letter 0]

21 Mar 2021

Response to Editor’s and reviewers’ comments

We thank the Editor and the two reviewers for detailed comments and suggestions on the manuscript. These suggestions have indeed helped improve the manuscript significantly. We present below a detailed point-wise response to the comments.

I. Response to comments by Editor

E1: PONE-D-20-35461

Retroactivity induced operating regime transition in a phosphorylation-dephosphorylation reaction cycle PLOS ONE Dear Dr. Viswanathan, Thank you for submitting your manuscript to PLOS ONE. After careful consideration, we feel that it has merit but does not fully meet PLOS ONE’s publication criteria as it currently stands. Therefore, we invite you to submit a revised version of the manuscript that addresses the points raised during the review process.

E2: Please include the following items when submitting your revised manuscript:

Response: We now submit these three documents.

E3: Response: We have now included the revised statement (in the response to comment E5 below) in the cover letter. Further, we have adhered to the guidelines corresponding to the figure files.

E4: When submitting your revision, we need you to address these additional requirements.

Response: We have now ensured that the manuscript meets the specified style.

E5: 2.Thank you for stating the following in the Acknowledgments Section of your manuscript:

"We thank Department of Science and Technology, 390 Government of India for funding this study."

Response: We have now deleted the statement regarding the funding information from the Acknowledgements section. We would like the funding statement to be modified as

"This study was supported by Science and Engineering Research Board, Department of Science and Technology, Government of India (MTR/2020/000589 and CRG/2020/002672) for funding this study. The funders had no role in study design, data collection and analysis, decision to publish, or preparation of the manuscript.”

We have now specified this modified funding statement in the cover letter with a request to amend the statements in the relevant places.

II. Response to Reviewers' comments

Q1. Is the manuscript technically sound, and do the data support the conclusions?

Reviewer #1: Partly

Reviewer #2: Partly

Response: We thank the reviewer for the opinion. Based on the comments from the reviewers, we have now revised the manuscript (and the supplementary material) extensively to tie tightly the conclusions drawn to the results presented in various figures.

Q2. Has the statistical analysis been performed appropriately and rigorously?

Reviewer #1: N/A

Reviewer #2: N/A

Response: N/A

Q3. Have the authors made all data underlying the findings in their manuscript fully available?

Reviewer #1: Yes

Reviewer #2: Yes

Response: We thank the reviewers for this observation.

Q4. Is the manuscript presented in an intelligible fashion and written in standard English?

Reviewer #1: No

Reviewer #2: Yes

Response: We thank the reviewer for highlighting in their comments various aspects that could be improved. We have now incorporated these and have improved clarity of presentation of the results, of conclusions drawn and of the discussions made in several sections. We have further fixed several grammatical, style-related, and typographical errors in the entire manuscript. We believe these changes would address the above concerns.

Q5. Review Comments to the Author

Response: We present below point-wise response to the comments and also refer to the manuscript wherever relevant to highlight the changes incorporated. 

R1.1 Reviewer #1: The study presented by Parundekar and Viswanathan investigates how sequestering the substrate or product of a phosphorylation/dephosphorylation cycle can shape the steady-state dose-response curve of the cycle, where the “dose” or input of the system corresponds to the concentration of the kinase catalyzing the phosphorylation. As a starting point, the authors derived an analytical expression for the steady-state phosphorylation of a simple model of phosphorylation/dephosphorylation including the sequestration of the substrate and/or product. The model output is studied by applying a classification scheme introduced by Gomez-Uribe et al. which categorizes dose-response curves into one of four characteristic operating-regimes such as hyperbolic or ultrasensitive. The main finding of the study is that protein sequestration can alter the shape of the dose-response curve and that by modifying the strength of sequestration, a cell could cause the dose-response curve of the phosphorylation/dephosphorylation cycle to shift from one operating regime into a different one.

Phosphorylation/dephosphorylation cycles are important and ubiqituous signaling motifs and the presented findings are interesting. 

Response: We thank the reviewer for the observations about the manuscript and valuable comments.

R1.2: Although the chosen approach is in principle rigorous, the current implementation seems to be flawed at several points. Please see the list of major and minor concerns below, many of which should be easy to address.

Response: We thank the reviewer for noting the rigor adopted in the work. We thank the reviewer for these comments and we have addressed these and incorporated them in detail in the revised version of the manuscript.

R1.3: Major issues:

(1) Most importantly: the derivation of equation [5] presented by the authors is, I believe, incorrect and some of the definitions for composite quantities provided by the authors are not consistent with equation [5]. Please refer to the attached document for further details and see whether you agree with my analysis. Generally, I think the derivation presented in appendix 1 would benefit from showing the full mass-action reaction scheme and smaller steps in the derivation

Response: We thank the reviewer for these observations, for concerns regarding the model employed in the study, and also the derivations attached. We agree that we did not present some of the underlying assumptions made and the derivation of the quasi-steady state approximation (QSSA) model. Specifically, we did not explicitly present the full ODE model. We believe these lacunae have caused the lack of clarity. As has also been suggested by Reviewer 2 in comment R2.2, we now present in Appendix I, in pages 22-23, lines (510-537) in the revised manuscript a detailed full model with the assumptions made. In pages 23-24, lines (538-553) in the revised manuscript, we present a detailed derivation of the QSSA model with intermediate steps. We also present definitions of the each of the quantities used. Subsequently, in 

R1.4: (2) The authors introduce the four operating regimes as defined by Gomez-Uribe with almost no explanation. Since many biologists or biochemists are not even familiar with the term ultrasensitivity, I strongly suggest to add a bit more detail here: How are these regimes defined? How do their dose-response curves look like? (A small figure similar to Fig 2 in Gomez-Uribe et al. 2007 would be helpful.) What are their properties, how does this translate into different signaling functions and why is this biologically important?

Response: We thank the reviewer for the suggestion to include a figure on the dose-response curves and the different operating regimes. We have now included in Fig. 2 representative dose-response curves of the four operating regimes. We have explicitly stated in Table 1 the conditions employed for drawing these curves and have discussed the associated definitions in detail in page 7, in lines (159-171)in the revised manuscript. Further, in lines (172-179) in the revised manuscript, we have discussed briefly the properties and potential biological relevance of the operating regimes along with citations.

R1.5: (3) It is not clear to me how the regime identification procedure based on relative distance/error can avoid misclassification. For instance, would a TH curve not be misclassified as ultrasensitive if the threshold coincides with the EC50 of the nominal US profile? (see attached picture: TH* is closer to US than to TH). 

Response: We thank the reviewer for bringing to our attention this possible lack of clarity. As shown in Fig. 4, the boundaries obtained using the metric adopted show that a clear separation between all regimes except Hyperbolic (H) and Ultrasensitive (U) which have a marginal overlap. We compare in Fig. R1.1 below the EC50 distribution of all the dose-response curves that were classified into the Ultrasensitive (U) and Threshold-Hyperbolic (TH) regimes. The distribution clearly shows that the dose-response curve that we classify into U regime cannot have an EC50 beyond that of those in TH regime in the parameter range considered. Moreover, since the metric in Eq. 8 (page 11, line 242 in the revised manuscript) uses a self-learned scaling and that ensures that misclassification can be predominantly circumvented. Note that the self-learned scaling is system parameter independent, unlike the metric adopted by Gomez-Uribe et al [Gomez-Uribe et al., PLoS Comp Biol., 2007], due to which the operating regimes reported had overlap among all the regimes. We have highlighted and discussed the importance of the metric used in this study in page 11, lines 245-251 in the revised manuscript.

<<figure in the attached 'Response to reviewers.docx' file>>

Figure R1.1: Boxplot showing the distribution of the EC50 for the dose-response curves in the Ultrasensitive and Threshold Hyperbolic regimes.

R1.6: Also, why is the area of the regimes (especially TH) in the K1 vs K2 plot so different from Gomez-Uribe et al. 2007?

Response: K1 vs K2 plot presented in Fig. 4 is different from Gomez-Uribe et al. because their model employs total quasi-steady state approximation which permits validity of the reduced model for a larger range of parameters. Further, their metric uses total substrate concentration as a metric and therefore their classification is strongly system parameter dependent. 

R1.7: (4) Since a regime transition due to sequestration has already been described before (Ventura et al.), the manuscript could benefit from extending the scope a little. Given that Gomez-Uribe et al. 2007 found the different regimes to influence low-pass filtering properties, I suspect studying the influence of k_on or k_off of the sequestration reactions on cycle dynamics and low-pass filtering could offer interesting insights which may increase the impact of the study (e.g. simple numerical analysis with the mass-action model).

Response: The reviewer is correct in observing that Ventura et al. has reported modulation of dose-response curves by sequestration. The Hill coefficient and S50, an alternative for EC50 pattern were used to make an observation of a couple of transitions via a few examples. However, this approach does not offer any insight into underlying causal mechanism governing such a modulation. Moreover, the systematic transition from different types of response are not considered either. 

The goal of this study is to systematically characterize operating regimes in the presence of sequestration, identify all possible transitions and the specific causal mechanisms governing these. This is stated in the introduction in the last paragraph in page 4 (lines 97-104) in the revised manuscript and also in line 216, page 10 in the revised manuscript. Different responses of the enzymatic futile cycle is governed by the levels of the saturation or unsaturation of the two enzymatic reactions. We contrast a dose-response curve with the nominal curves, which incorporates the saturated and unsaturated states of the enzymes, using a metric and characterize them systematically. To the best of our knowledge, our study is the first to systematically characterize all four operating regimes of a futile cycle in the presence of retroactivity (Fig. 4 of the revised manuscript), and also the 12 transitions between them. Further we identify the precise causal mechanism governing retroactivity driven regime transition (Figs. 5 and 6)

We agree with the reviewer that effect of sequestration parameters on the dynamics and low-pass filtering properties are indeed important and will offer new insights into the dynamical behaviour of the enzymatic futile cycle. However, we would like to restrict this study to only the steady-state response of the cycle and we think analysing the dynamical behaviour is indeed a logical extension of this study which we will consider in a separate manuscript in the future.

R1.8: Minor issues: (1) Abstract, p2. Line 35: I don’t think “sequestration strength of 0.3” is a helpful description for most readers. A qualitative description would be more helpful.

Response: We thank the reviewer for bring this to our attention. Both in abstract and also in section 3, page 13, lines 298-303, we have now provided a logical interpretation of the “sequestration strength of 0.3” which we believe would help readers appreciate the importance of it better.

R1.9: (2) The expression “PdPC” seems a bit cumbersome to me. Since the studied motif not only applies to phosphorylation/dephosphorylation cycles but also to other PTMs or to GTPase cycles, I find it more appropriate to use a more common and general expression such as activation/inactivation cycle, PTM cycle, futile cycle …

Response: We have now replaced PdPC with ‘enzymatic futile cycle’ everywhere in the manuscript, including the title.

R1.10: (3) I don’t understand how increasing the input dose (e.g. kinase concentration) of a cycle can itself induce a regime shift? (p.4, line 77)

Response: We thank the reviewer to bringing to our attention this lack of clarity. The input here refers to the stimulus that triggers the cells and thereby activates the MAPK cascade. We inadvertently referred to this as “dose-strength dependent”. For better clarity and capture of the findings reported in ref. [21], we have now split and re-phrased the entire sentence into two sentences in pages 3-4, lines 75-79.

R1.11: (4) p.5, line 118-119: the authors speak of a QSSA for MS1 and MS2, yet I see no use of MS1 or MS2 QSSA in the whole paper

Response: We agree with the reviewer. We have now re-phrased the sentence in page 6, line124-125.

R1.12: (5) Where does equation [7] come from? From the cited references or from the authors’ study solved manually/by computer algebra system?

Response: To get Eq. (7), we solved the quadratic equation obtained by setting the lhs to zero in Eq. (5). We have now explicitly stated so in page 6, lines145-146. 

R1.13: (6) Figure 2: The plot of the nominal regime ST almost overlaps with the Y axis and upper border of the figure. The characteristic feature of the ST regime (linear increase until saturation) cannot be identified here! Better rescale the axis or use log(X) axis.

Response: We agree with the reviewer that the linear increase property of the ST regime was unclear in Fig. 2 of the first submitted version of the manuscript. In the revised version of the manuscript, in Fig. 3 -- previously Fig. 2 --- we have provided a zoomed version of both the plots to show the linear increase until saturation for the nominal profile of ST. Morever, we have described the same in the caption for the figure.

R1.14: (7) The rate balance figures (figure 4 and supplementary figures) are quite crowded and it would help to use color gradient lines (whose numerical values are given in a legend) instead of all the arrows and values. For figure 4 in particular, it would be very helpful to also show 3D dose-response plots visualising the regime transitions (Z-axis = m, X-axis = e_t, Y-axis = α or λ). These are often more intuitive to understand than rate balance plots.

Response: We agree with the reviewer that Fig. 4 and the last two figures in Text S1 were quite crowded. As suggested, we have now provided a colorbar (with numerical values specifying the legend) for the each of the panels in Fig. 5 and also in the last two figures in Text S1. Moreover, we have also included an additional figure for every panel in Figure 5 capturing the dependence of dose-response curves on the retroactivity strengths used. Moreover, such a plot corresponding to the last two figures in Text S1 is now presented in Fig 6, which are discussed in a new sub-section 3.3.4, as suggested by Reviewer 2 in R2.9.

Further, in order to help a reader understand the results from the rate-balance plots better and correlate the inferences with the dose-response curves and sensitivity, we have extensively rephrased the corresponding description of results in sub-sections 3.3.1-3.3.4.

R1.15: (8) Sections 3.3.1 and 3.3.2 are largely descriptive of the figures and could be shortened by focussing more on the overall effect of changing α or λ at a nominal operating point and by moving parameter values to the figure legends.

Response: We have now re-written the entire section and wherever possible, we have removed redundant descriptions and focussed on the overall effects captured by the figures. We have also included helpful descriptions and some parameter values in the figure captions.

R2.1 Reviewer #2: In this paper, the authors investigate the operating regimes of a signaling cycle consisting of a protein that can be in an inactive or active form: the protein is activated and deactivated by two enzymatic species, a kinase and phosphatase, respectively. The main novelty introduced in this paper is the presence of a downstream load on the protein (on both the forms) determining retroactivity. The authors characterize the effect of retroactivity on the input-output relationship (i.e. operating regime) of the cycle, where the input is the kinase concentration and the output the active form of the protein (normalized with respect to its total concentration). Moreover, they find that increasing retroactivity strength can trigger five possible regime transitions: for four possible transitions, they show that the modulation of the saturation levels of the enzymatic reactions by increasing retroactivity is the main reason for the operating regime transitions.

The paper is interesting. I have the following comments/questions.

Response: We thank the reviewer for the observations and the comments.

R2.2: The quasi-steady-state approximation (QSSA) is employed to study the system defined by biochemical reactions [1-4] and get Eq. [5]. However, the full ODE model should also be implemented and simulated: the corresponding steady-state results should be compared with those obtained by exploiting QSSA in order to verify and validate this approximation.

Response: We thank the reviewer for this suggestion. In Text S1.1, we present a detailed solution of the full ODE model at steady-state conditions. The full ODE model is presented in Appendix I in the revised version of the manuscript as suggested by Reviewer 1 in R1.3. The solution of the full model matches with the solution of the QSSA model in Eq. (5) in the manuscript.

R2.3: About Fig. 2, it would be better to use inserts in panels A and B showing a zoom-in of the active form of protein vs the kinase concentration (e_t) on lower e_t values.

Moreover, be sure to use a small step size for lower e_t values in order to not miss any significant behavior and confirm the obtained results.

Response: We thank the reviewer to bring this point to our attention. In the revised manuscript, we have now incorporated a zoomed version of the two sub-figures as inset in Fig. 3 (which was Fig. 2 in the version submitted earlier). The zoomed version now shows the linear increase property of the ST nominal profile.

R2.4: From this study, it seems to emerge that the ultrasensitive threshold (i.e. EC50) can be modulated by retroactivity. Could the authors give more details, in particular, by quantifying the modulation?

Response: We thank the reviewer for this suggestion of quantifying the EC50 in the Ultrasensitive regime. We capture the S50 (Eq. 11 in the revised manuscript), which was used to estimate the EC50, for different dose-response curves at different retroactivity strengths. We now present the effect of retroactivity strength on the EC50 in Fig. S1.3 in the Suppl Text S1. We briefly discuss this effect in sub-sections 3.3.1 and 3.3.2 in page 14, lines 332-334 and in page 17, 386-387. 

R2.5: Could the authors give more details about the criterion defined by Eq. 8?

Response: We thank the reviewer for suggesting to provide more details on Eq. 8. We now present a detailed discussion on the origin and basis for the criterion and also the advantages of the same in page 11, lines 243-251 in the revised version of the manuscript.

R2.6: For each figure, the caption should be more exhaustive.

I would recommend to the authors to improve understanding/readability of the manuscript to provide more details in the figure captions.

Response: We thank the reviewer for this suggestion. We agree that certain details are missing in the figure captions. We have now revised the captions all the existing and new figures both in the main text and supplementary information. We provide detailed description and details in each of the captions.

R2.7:L Could the authors explain better the results reported in Section 3.3 and shown in Fig.4? 

Response: We thank the reviewer for this suggestion. We have now re-written the entire subsection 3.3 corresponding to Fig 5 (which is Fig. 4 in the originally submitted manuscript) with better explanation. We have also introduced a dose-response curves figure corresponding to each of the transitions to explain the results better, as suggested by reviewer 1 in R1.14. 

R2.8: Is the dotted line reported in each panel representing the transition from a regime to another (from U to TH for panels A and B and from U to ST for panels C and D)? 

Response: We thank the reviewer for bringing this lack of clarity to our attention. Yes, dotted line represents the transition. We have now explicitly stated that dotted line corresponds to the transition in the figure caption, in page 16, line 380 in the revised version of the manuscript.

R2.9: Could the authors explain better in the text how retroactivity strength modulates the saturation levels of the enzymatic reactions?

Response: We thank the reviewer for this concern. We have now explained in the revised manuscript in page 15, in the lines 348-352: “The nature of an enzymatic reaction being saturated, that is, all enzymes bound to its substrate, is specified by the range of m̅ for which the corresponding rate does not change significantly. Therefore a sufficient proportional increase in K̅1(λ) due to λ can lead to Rp exhibiting a linear dependence on m̅ in a certain range. The nature of the phosphorylation reaction is unsaturated in this range of m̅.”

R2.10: Instead of summarizing the results obtained for the other transitions at the end of Section 3.3.2, it would be better to add another subsection and provide more details of the results reported in the supplementary text.

Response: We thank the reviewer for this suggestion. We have now moved the H to ST transition into Fig. 5 (previously Fig 4). Further, as suggested by Reviewer 1 in R1.14, we have introduced the dose-response curves in Figs. 5A, 5D, 5G and in Fig. 6 corresponding to the five transitions explained. We have now re-written the sub-sections 3.3.1 and 3.3.2 and also introduced a new sections 3.3.3 and 3.3.4. In these four sub-sections, we analyse and identify the causal mechanisms governing these 5 transitions.

R2.11: Minor comments In Section 2, when the enzyme concentrations e_t and p_t are defined, it should be specified that these concentrations represent the total concentrations of E and P, respectively, in the unbounded and bounded forms.

Response: We thank the reviewer for bringing this issue to our attention. We have now include the word “total” in page 6, line130-131.

R2.12: Please check punctuation, as line 210 (it is missing a full stop).

Response: We have now thoroughly checked the entire manuscript and fixed several grammatical and typographical errors.

R2.13: Define the abbreviations (as rhs) at first occurrence.

Response: We have now defined abbreviations at first occurrence. For example, “right hand side” in page 6, line 142.

R2.14: Line 273, it should be et=3000 nM in Rp.

Response: We have incorporated this correction in line 428, page 18 in the revised manuscript. 

R2.15: Line 275, it should be lambda=0 in Rp.

Response: We have incorporated this correction in line 355, page 15 in the revised manuscript. We have now thoroughly checked the numbers and symbols and their consistency throughout the revised manuscript.

R2.16: Line 278, falls instead of fall.

Response: We have re-phrased this sentence. Further we have thoroughly revised the manuscript and fixed several typographical and grammatical errors.

6. PLOS authors have the option to publish the peer review history of their article (what does this mean?). If published, this will include your full peer review and any attached files.

Do you want your identity to be public for this peer review? For information about this choice, including consent withdrawal, please see our Privacy Policy.

Reviewer #1: Yes: Daniel Koch

Reviewer #2: No

---

## [Decision Letter · Decision Letter 1]

15 Apr 2021

Retroactivity induced operating regime transition in an enzymatic futile cycle

PONE-D-20-35461R1

Dear Dr. Viswanathan,

We’re pleased to inform you that your manuscript has been judged scientifically suitable for publication and will be formally accepted for publication once it meets all outstanding technical requirements.

Kind regards,

Christopher Rao

Academic Editor

PLOS ONE

Additional Editor Comments (optional):

Reviewers' comments:

Reviewer's Responses to Questions

**Comments to the Author**

1. If the authors have adequately addressed your comments raised in a previous round of review and you feel that this manuscript is now acceptable for publication, you may indicate that here to bypass the “Comments to the Author” section, enter your conflict of interest statement in the “Confidential to Editor” section, and submit your "Accept" recommendation.

Reviewer #1: All comments have been addressed

Reviewer #2: All comments have been addressed

2. Is the manuscript technically sound, and do the data support the conclusions?

Reviewer #1: Yes

Reviewer #2: Yes

3. Has the statistical analysis been performed appropriately and rigorously? 

Reviewer #1: Yes

Reviewer #2: N/A

4. Have the authors made all data underlying the findings in their manuscript fully available?

Reviewer #1: Yes

Reviewer #2: Yes

5. Is the manuscript presented in an intelligible fashion and written in standard English?

Reviewer #1: Yes

Reviewer #2: Yes

6. Review Comments to the Author

Reviewer #1: In my view, the authors have sufficiently addressed the raised issues and the overall clarity of the presentation and figures improved notably. A last comment: I might be wrong since I'm not a native speaker, but I feel there are still some language issues, e.g. with the first sentence of the abstract: "Activated phosphorylation-dephosphorylation biochemical reaction cycle is a class of enzymatic futile cycles." Should it not be "THE activated phosphorylation-dephosphorylation biochemical reaction cycle (...)" or "Activated phosphorylation-dephosphorylation biochemical reaction cycleS ARE (...)"?

However, I see no reason to further delay publication of the authors' interesting findings and trust they will give the manuscript another round of proof-reading before publication.

Reviewer #2: The authors have addressed all my previous comments.

I have few minor comments:

Line 143 of the 'Revised Manuscript with Track Changes', please define K1 and K2.

Line 624, it should be PMp.

For fig 3 (and for similar figs), please add in the caption the K1_bar and K2_bar values for obtaining the nominal U profile (blu dotted curve, shown also in fig 2) as done for the yellow curve.

7. PLOS authors have the option to publish the peer review history of their article (what does this mean?). If published, this will include your full peer review and any attached files.

Reviewer #1: **Yes: **Daniel Koch

Reviewer #2: No

---

## [Editor Report · Acceptance letter]

22 Apr 2021

PONE-D-20-35461R1 

Retroactivity induced operating regime transition in an enzymatic futile cycle 

Dear Dr. Viswanathan:

I'm pleased to inform you that your manuscript has been deemed suitable for publication in PLOS ONE. Congratulations! Your manuscript is now with our production department. 

Kind regards, 

on behalf of

Dr. Christopher Rao 

Academic Editor

PLOS ONE